# PEEK Biomaterial in Long-Term Provisional Implant Restorations: A Review

**DOI:** 10.3390/jfb13020033

**Published:** 2022-03-22

**Authors:** Suphachai Suphangul, Dinesh Rokaya, Chatruethai Kanchanasobhana, Pimduen Rungsiyakull, Pisaisit Chaijareenont

**Affiliations:** 1Department of Prosthodontics, Faculty of Dentistry, Chiang Mai University, Chiang Mai 50200, Thailand; kungomfs@gmail.com (S.S.); chatruethai.kan@gmail.com (C.K.); pimduen.rungsiyakull@cmu.ac.th (P.R.); 2Department of Clinical Dentistry, Walailak University International College of Dentistry, Walailak University, Bangkok 10400, Thailand; dinesh.ro@wu.ac.th

**Keywords:** polyetheretherketone, PEEK, carbon fiber, provisional restoration, provisional crown, implant prostheses, implant abutment

## Abstract

Polyetheretherketone (PEEK) has become a useful polymeric biomaterial due to its superior properties and has been increasingly used in dentistry, especially in prosthetic dentistry and dental implantology. Promising applications of PEEK in dentistry are dental implants, temporary abutment, implant-supported provisional crowns, fixed prosthesis, removable denture framework, and finger prosthesis. PEEK as a long-term provisional implant restoration has not been studied much. Hence, this review article aims to review PEEK as a long-term provisional implant restoration for applications focusing on implant dentistry. Articles published in English on PEEK biomaterial for long-term provisional implant restoration were searched in Google Scholar, ScienceDirect, PubMed/MEDLINE, and Scopus. Then, relevant articles were selected and included in this literature review. PEEK presents suitable properties for various implant components in implant dentistry, including temporary and long-term provisional restorations. The modifications of PEEK result in wider applications in clinical dentistry. The PEEK reinforced by 30–50% carbon fibers can be a suitable material for the various implant components in dentistry.

## 1. Introduction

Currently, digital technologies have been integrated into implant dentistry for prosthetic restoration rehabilitation of missing teeth [1,2,3,4]. For implant components, titanium (Ti) and its alloys are commonly used but they have disadvantages of causing black coloration in the anterior zone and causing allergy [5,6]. Recently there have been tried thermoplastic materials in implant dentistry. Polymers are one of the most important materials in dentistry. They have present good physical and mechanical properties and biocompatibility. Denture base, removable appliances, and restorations are fabricated mainly from polymers [7]. In addition to these, various polymeric thin films are also available for various dental applications [8].

The polyaryletherketone (PAEK) family are superior performance polymers among all thermoplastic composites, as shown in Figure 1 [9,10]. Polyetheretherketone (PEEK) and polyetherketoneketone (PEKK) are the two most common sub-members of the PAEK family launched in the engineering field in the 1980s [11]. PAEK family has a linear aromatic polyether ketone. PEEK presents ultrahigh performance with superior properties and chemical resistance [12,13]. These remarkable outcomes of PEEK caused it to receive much attention in the field of medicine and dentistry [14,15,16,17,18,19].

Recently, although PEEK has been increasingly used in implant dentistry, PEEK as a long-term provisional implant restoration has not been studied extensively. Hence, this review article aims to review PEEK as a long-term provisional implant restoration for applications in dentistry. Articles published in English on PEEK biomaterial for long-term provisional implant restoration were searched in Google Scholar, ScienceDirect, PubMed/MEDLINE, and Scopus. Then, relevant articles were selected and included in this literature review.

## 2. PEEK: Structure, Properties, and Prosthetic Applications

PEEK was developed in the late 1990s as a biomaterial that has superior physical, mechanical, and biological properties for biomedical applications [18]. It is a thermoplastic, monochromatic, semicrystalline polymer and is considered the most important member of the PEAK family [20]. The structure of PEEK comprises repeated aromatic rings of the ether and ketone groups (Figure 2) [21].

The high-performance PEEK polymer is being used in the field of orthopedics, spinal, and dental prostheses [18,22]. Recently, PEEK has been increasingly used in implant dentistry. Promising applications of PEEK in dentistry include dental implants [12,23], temporary abutments [22], implant-supported provisional crowns [24,25], fixed prostheses [21], removable denture frameworks [17], and finger prostheses [26]. Little information is available in the dental literature on PEEK as a long-term provisional implant restoration [14,15]. Hence, this review article aims to review PEEK as a long-term provisional implant restoration for applications in dentistry.

A three-dimensional (3D) finite element analysis of implants, abutments, screws, crowns, and peri-implant bones showed that using PEEK materials such as PEEK abutments and crowns reduces the stresses on abutment [27]. In addition, a PEEK framework veneered with composite resin has an elastic modulus of around 4 GPa, which can dampen the occlusal forces on tooth structures, thus protecting structures better than ceramic materials and preventing debonding rates [16,25,28]. Hence, they can be used as alternative materials in endocrown for severely damaged molar and implant restorations.

PEEK has some advantages in dentistry compared with metallic restorations such as Ti. Ti is an allergen, whereas PEEK causes fewer hypersensitive and allergic reactions [5,6]. In addition, PEEK has a more aesthetic appearance than Ti, and PEEK has less/no effect on magnetic resonance imaging artifacts, as it is radiolucent [29,30]. Hence, PEEK abutments can be used as alternatives to Ti abutments.

Recently, studies have focused on improving the surface of PEEK, the bioactivity of PEEK implants at the nanoscale, and reducing the biofilm formation in PEEK materials in implant dentistry [12,23,31,32]. Surface modification of PEEK enhances the cell adhesion, proliferation, biocompatibility, and osteogenic properties of PEEK implant materials. PEEK also influences the biofilm structure and reduces the chances of peri-implant inflammations [31].

A scan body is being manufactured by various companies and varies according to the material, size, shape, connection type, reusability, scanner compatibility, computer design software, and cost [33,34]. Various factors of the scan body (material, design, light reflectance, manufacturing tolerance, and screwing) affect digital transfer in implants [35,36,37,38,39,40]. PEEK scan bodies have been used in prosthetic dentistry using digital technologies. Digital technologies have increased clinical success [41,42], and digital impressions are more accurate and time-efficient than conventional impressions [33,42,43,44]. Still, there can be PEEK displacements in PEEK scan bodies, especially with hand tightening [35]. Figure 3 shows the vertical displacements of PEEK scan bodies (>100 µm) (Myfit and Dentium). Hence, it is suggested to be tightening torque (5 Ncm) but not hand tightening [35].

## 3. Provisional Restorations

PEEK is a suitable material for provisional restorations, including fixed dental prostheses during dental implant treatment [14,16,25]. Hexagonal-shaped healing abutments can help to provide adequate retention and resistance for implant prostheses [14]. PEEK biomaterials are becoming popular as healing abutments and in PEEK with titanium base temporary abutments. Figure 4 shows the various PEEK abutments. PEEK healing abutment (Figure 4B) is used as an alternative for classic titanium healing abutment, and PEEK with titanium base temporary abutment (Figure 4C) is used for long-term interim restoration, especially in areas with esthetic consideration.

## 4. A Case of PEEK Temporary Abutment Fabrication

The use of a PEKK temporary abutment (RC Temporary Abutment, Straumann, Basel, Switzerland) as a provisional restoration is shown in Figure 5. Following implant placement, a PEKK temporary abutment try-in is performed, and the temporary abutment is adjusted until desired shape and size were achieved. Then, a temporary crown (Protemp 4, 3M ESPE, Seefeld, Germany) is attached to the customized PEEK abutment, after which it can be inserted into the patient. The patient can use it for a few weeks to months. Then, after some months, a definitive implant prosthesis can be fabricated and placed in the patient.

In a study by Santing et al. [24], the authors measured the failure types and fracture strength of implant-supported PEEK abutments with crowns made from composite crowns and solid Ti temporary abutments and found no difference between different abutment types. In addition, the fracture strength of temporary crowns on PEEK abutments was similar to the Ti temporary abutments except for the central incisors. In addition, Stawarczyk et al. [45] studied the effect of pretreatments of PEEK and adhesive systems on the fracture load of PEEK frameworks for fixed dental prostheses (FDPs). They found that all studied FDPs showed some cracks in the veneering composite crowns in the pontic region irrespective of the pretreatment or adhesive used. In addition, the PEEK frameworks did not fracture after loading, but chipping of the composite was found (Figure 6).

In a study by Rodríguez et al. [46], the authors studied the fracture pattern and fracture load of frameworks for FPDs made with three different subtractive computers aided design and computer-aided manufacturing (CAD/CAM) materials (milled metal made from cobalt–chromium, PEEK, and Lava zirconia), and they found that PEEK had a higher fracture load than zirconia but a lower one than milled metal, as shown in Figure 7. All groups showed clinically acceptable fracture load (>1000 N). Hence, PEEK might be considered a favorable alternative for FPDs.

## 5. PEEK Modifications

In screw-retained implant prostheses, a PEEK abutment screw has an advantage over a metal screw due to similar elastic properties [47]. In addition, the stress distribution in implant and abutment can be improved by composing the PEEK and its fiber-reinforced composites [47,48,49,50,51]. Figure 8 shows the surfaces of a 3D-printed PEEK and PEEK matrix composite [52]. The unfilled PEEK has low elastic modulus and strength. It has been found that addition of 30–50% carbon fibers does not affect the stress compared with unfilled PEEK [48], but >50% carbon fibers show higher elastic modulus and strength than those of the PEEK composites with lower percentage of carbon fibers [47]. Increasing short carbon fibers (CFs) up to 60% results in higher stress distribution through abutment and implants toward the bone [48]. Hence, the strength and stiffness of PEEK composites can be enhanced by adding more carbon fiber networks. Furthermore, PEEK veneered onto Ti structures can also enhance the stress distribution, especially at the interface between the implant and bone [48].

Similarly, Neumann et al. [49] compared the fracture resistance of Ti, PEEK, and PEEK reinforced with 30% carbon fiber abutment screws, and they found that the Ti screws had higher fracture resistance than screws with PEEK and PEEK reinforced with 30% carbon fiber but no difference between screws with PEEK and PEEK reinforced with 30% carbon fiber. In addition, Schwitalla et al. [47] conducted a study on PEEK screw-in screw-retained implant prostheses, in which they used four PEEK-modified abutments: two screw types with 15% short carbon fibers and 40% short carbon, and another two screw types with 20% TiO_2_ powder and >50% continuous parallel carbon fibers. Controls included Ti6Al4V abutments screws. They found that the maximum tensile strength was 76.08 MPa for 20% TiO_2_ powder, 152.67 MPa for 15% short carbon fibers, 157.29 MPa for 40% short carbon fibers, and 191.69 MPa for 15% continuous carbon fibers. The maximum tensile strength of the Ti screws amounted to 1196.29 MPa. The PEEK screw could be removed more easily in case of abutment screw fracture. Hence, the PEEK reinforced >50% continuous carbon fibers could be a suitable material to use in implant dentistry.

In another study, Schwitalla et al. [50] studied the insertion torque at the failure of PEEK using implants made from unfilled PEEK, PEEK-reinforced carbon fiber (>50 vol% continuous axially parallel fibers), and Ti. The insertion torque at the failure of the unfilled PEEK implants was 22.6 Ncm, which was higher than that of the PEEK-reinforced carbon fiber (20.2 Ncm) but significantly less than that of the Ti specimens (92.6 ± 2.3 Ncm). Hence, the PEEK and PEEK-reinforced carbon fiber implants are not able to withstand the insertion force necessary to obtain primary stability for immediate loading.

Although the PEEK has good aesthetics and mechanical properties similar to the human bone, PEEK materials are at risk of fracture and abrasion [52]. Further studies are needed on PEEK implants so that they can replace titanium in the future [31,53,54]. In addition, it is necessary to improve the fracture properties of PEEK implants. Considering this, the development of a PEEK composite by adding continuous multidirectional carbon fibers may help to improve the properties, but more studies are needed [51].

## 6. Conclusions

PEEK presents suitable properties for various implant components in implant dentistry, including temporary and long-term provisional restorations. Although PEEK has good aesthetics and mechanical properties, similar to the human bone, PEEK materials are at risk of fracture and abrasion. The modification of PEEK is necessary for its wider applications in dentistry. PEEK reinforced by 30–50% carbon fibers would be the material of choice for various dental implant components. Further studies are needed on PEEK implants so that they can replace Ti in the future and be used in long-term provisional restorations.

## Figures and Tables

**Figure 1 jfb-13-00033-f001:**
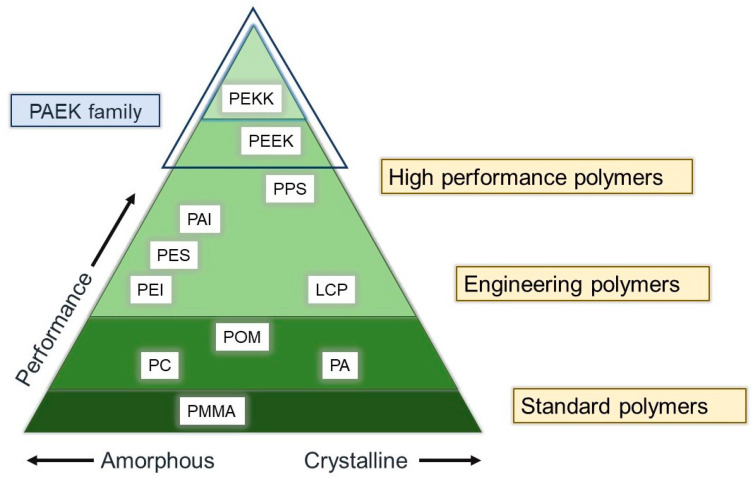
Various polymers and performance of PEKK: PMMA = polymethylmethacrylate, PC = polycarbonates, pA = polyamide, POM = polyoxymethylene, PPS = polyphenylene sulfide, PAI = polyamide-imide, PEI = polyethylenimine, PES = polyether sulfone, PEEK = polyetheretherketone, PEKK = polyetherketoneketone, PAEK = polyaryletherketone.

**Figure 2 jfb-13-00033-f002:**
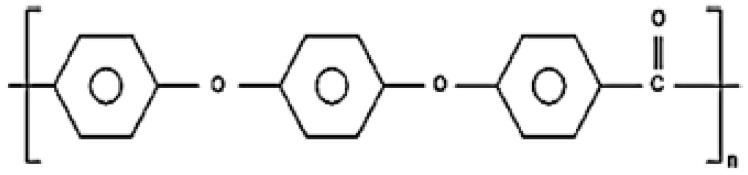
Structure of polyetheretherketone (PEEK).

**Figure 3 jfb-13-00033-f003:**
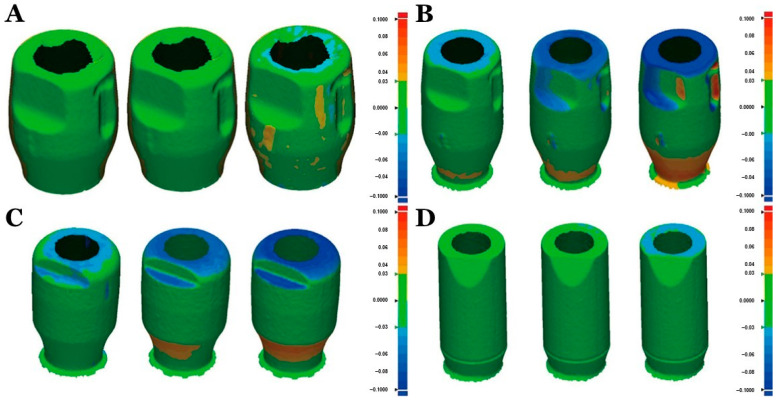
Various scan bodies (left to right: 5 Ncm, 10 Ncm, and hand tightening). Myfit group (metal) (**A**), Myfit (PEEK) (**B**), Dentium (PEEK) (**C**), and Straumann (PEEK) (**D**) [35].

**Figure 4 jfb-13-00033-f004:**
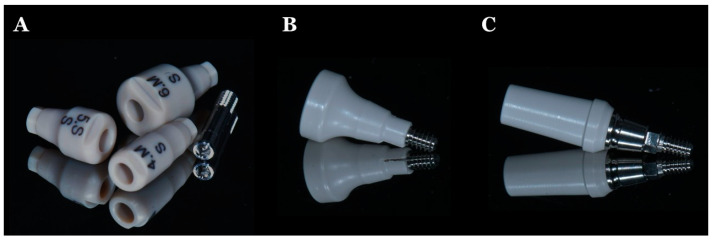
PEEK abutments: (**A**) PEEK scannable healing abutment (Dentium Co., Seoul, Korea); (**B**) PEEK healing abutment (GM Customizable healing abutment, Neodent, Curitiba, Brazil); (**C**) PEEK with titanium base temporary abutment (GM PRO PEEK Abutment, Neodent, Curitiba, Brazil).

**Figure 5 jfb-13-00033-f005:**
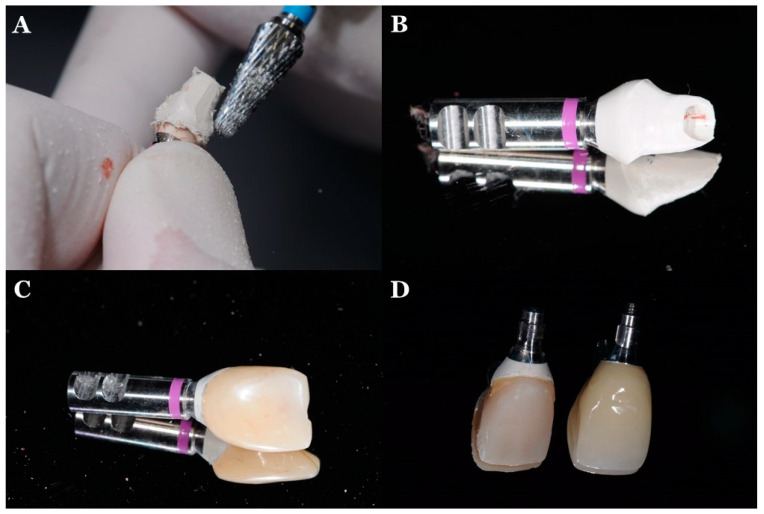
Fabrication of the PEEK with Ti base temporary abutment: (**A**) adjustment of the prefabricated PEEK temporary abutment; (**B**) customized PEEK abutment; (**C**) customized PEEK abutment with temporary crown attached to the abutment; (**D**) temporary implant prosthesis with PEEK abutment and definitive implant prosthesis with Ti abutment.

**Figure 6 jfb-13-00033-f006:**
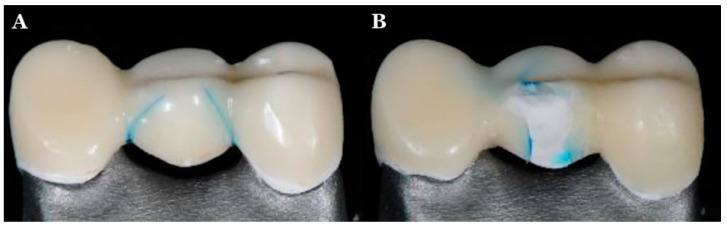
Failures of PEEK framework and composite crown: (**A**) cracks seen in veneering composite material following thermal cycling and (**B**) chipping of the veneering material seen after fracture testing [46].

**Figure 7 jfb-13-00033-f007:**
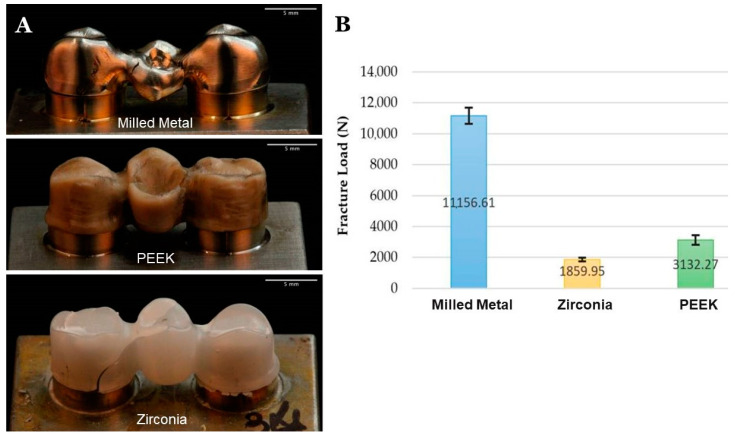
Fracture pattern (**A**) and fracture load (**B**) of metal, PEEK, and zirconia framework showed that PEEK had a higher fracture load than zirconia but a lower one than milled metal [46].

**Figure 8 jfb-13-00033-f008:**
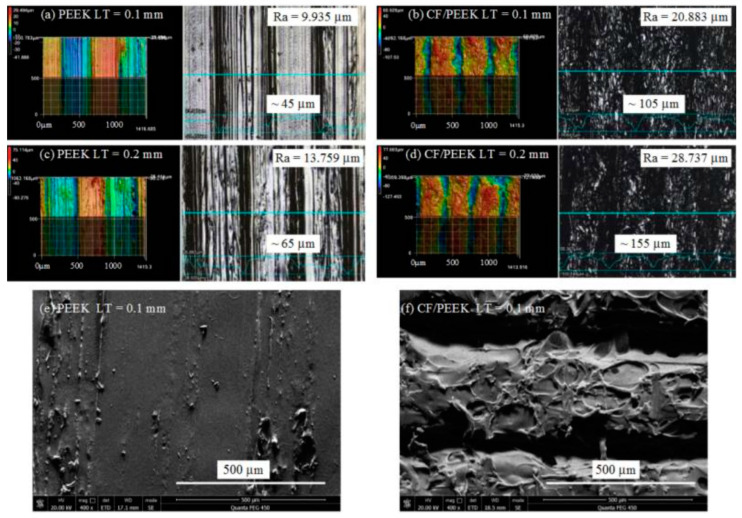
Surfaces 3D morphology and scanning electron microscopy of printed PEEK and Carbon fiber/PEEK composite: (**a**) PEEK with 0.1 layer thickness; (**b**) carbon fiber/PEEK with 0.1 mm layer thickness; (**c**) PEEK part with 0.2 mm layer thickness; (**d**) carbon fiber/PEEK part with LT 0.2 mm layer thickness; (**e**) scanning electron microscopy image of the PEEK part with 0.1 mm layer thickness; (**f**) scanning electron microscopy image of the carbon fiber/PEEK part with 0.1 mm layer thickness [51].

## Data Availability

Not applicable.

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
