# Peer review of "PEEK Biomaterial in Long-Term Provisional Implant Restorations: A Review"

_jfb, 2022, doi:10.3390/jfb13020033_

Round 1

Reviewer 1 Report

Suphangul et al present a review article that review the PEEK materials as a long-term provisional implant restoration for dental applications. They also provided case studies to support the description in the review. My feeling is that this is a good popular science article with small input of recent progress in the relavent area. Therefore, I suggest publication after English polishing. I strongly recommend that the author find an English polishing agent to edit English.

Author Response

Response to Reviewer 1 Comments

Suphangul et al present a review article that review the PEEK materials as a long-term provisional implant restoration for dental applications. They also provided case studies to support the description in the review. My feeling is that this is a good popular science article with small input of recent progress in the relavent area. Therefore, I suggest publication after English polishing. I strongly recommend that the author find an English polishing agent to edit English.

Thank you for your positive comments. For the case studies, the Managing Editor suggested adding only prosthesis pictures to remove the patient’s pictures due as we have not obtained ethical clearance.

The manuscript is edited and the English language is thoroughly improved.

Reviewer 2 Report

The manuscript is of interest and overall well-written. I would suggest the authors to address the following issues:

  • Materials and methods: reporting the terms used for the literature search and the time span of the included article is more suitable for a systematic review. Since the authors conducted a narrative review, specifying the search strategy is unnecessary.
  • The authors are suggested to reorganize the text in order to describe first the evidence from the literature, and then dedicate a specific section to case presentation and evidence from their institution. In particular, more information should be provided on the cases described. The authors are suggested to improve case presentation.

Author Response

Response to Reviewer 2 Comments

The manuscript is of interest and overall well-written. I would suggest the authors to address the following issues:

  • Materials and methods: reporting the terms used for the literature search and the time span of the included article is more suitable for a systematic review. Since the authors conducted a narrative review, specifying the search strategy is unnecessary.
  • Response: Thank you for your positive comments. Corrections in the Manuscript are highlighted in Yellow color. The literature search and time span are removed from the abstract and the manuscript.

  • The authors are suggested to reorganize the text in order to describe first the evidence from the literature, and then dedicate a specific section to case presentation and evidence from their institution. In particular, more information should be provided on the cases described. The authors are suggested to improve case presentation.
  • Response: The literature is organized with a separate section (Section 6) of the case presentation is added. There are limited resources on the PEEK as a provisional restoration. We did our best to add more relevant literature with references are added. More information is provided on the case. But for the case photos, the Managing Editor suggested us to add only prosthesis pictures and remove the patient’s pictures as we have not obtained ethical clearance.

Reviewer 3 Report

It is an interesting study, but need more evidence to making the point valid that PEEK is a suitable material

The background of the study is described. However, a limited number of references is cited to support the statements.

In abstract the key words and number of years have been mentioned whereas, in the literature it is not, so authors might want to mention it in review as well

Please check grammar and English through out the manuscript

Decide if it is a critical or literature review and use the same words throughout

Improve abstract’s ending and be specific

Don’t start a sentence with ‘And’

Introduce the abbreviations when they first appear in the text.

Add reference after sentence page 3 line 75

Add reference for figure 4,5

Concluded-page 6 line 174-spell check

Conclusion needs to be succinate, yet it should justify the aims of study, and it needs more input.

Author Response

Response to Reviewer 3 Comments

It is an interesting study, but need more evidence to making the point valid that PEEK is a suitable material

Thank you for your positive comments. Corrections are highlighted in Green color.

The background of the study is described. However, a limited number of references is cited to support the statements.

Response: There are limited resources on the PEEK as a provisional restoration. We did our best to add more relevant literature with references are added.

In abstract the key words and number of years have been mentioned whereas, in the literature it is not, so authors might want to mention it in review as well

Response: The literature search words and years are removed.

Please check grammar and English through out the manuscript

Response: The manuscript is edited and the English language is thoroughly improved.

Decide if it is a critical or literature review and use the same words throughout

Improve abstract’s ending and be specific

Response: We used the word literature review throughout the manuscript.

Don’t start a sentence with ‘And’

Response: The sentences starting with ‘And’ are edited.

Introduce the abbreviations when they first appear in the text.

Response: The abbreviations are introduced when they appear first throughout the manuscript.

Add reference after sentence page 3 line 75

Response: References are added on Page 3, line 75.

Add reference for figure 4,5

Response: Figure 4 and Figure 5 are our own.

Concluded-page 6 line 174-spell check

Response: Correction is done.

Conclusion needs to be succinate, yet it should justify the aims of study, and it needs more input.

Response: Conclusion is edited and improved.

Round 2

Reviewer 1 Report

I suggest publication.

Reviewer 2 Report

The authors addressed all the comments raised in the previous revision. There are no further comments.